# Resting-state fMRI signals contain spectral signatures of local hemodynamic response timing

Sydney M Bailes[1], Daniel EP Gomez[1,2,3], Beverly Setzer[1,4], Laura D Lewis[1,2,5,6]*

[1]Department of Biomedical Engineering, Boston University, Boston, United States; [2]Athinoula A. Martinos Center for Biomedical Imaging, Massachusetts General Hospital, Charlestown, United States; [3]Department of Radiology, Harvard Medical School, Boston, United States; [4]Graduate Program for Neuroscience, Boston University, Boston, United States; [5]Institute for Medical Engineering and Science, Massachusetts Institute of Technology, Cambridge, United States; [6]Department of Electrical Engineering and Computer Science, Massachusetts Institute of Technology, Cambridge, United States

*For correspondence:
ldlewis@mit.edu

Competing interest: The authors declare that no competing interests exist.

**Abstract** Functional magnetic resonance imaging (fMRI) has proven to be a powerful tool for noninvasively measuring human brain activity; yet, thus far, fMRI has been relatively limited in its temporal resolution. A key challenge is understanding the relationship between neural activity and the blood-oxygenation-level-dependent (BOLD) signal obtained from fMRI, generally modeled by the hemodynamic response function (HRF). The timing of the HRF varies across the brain and individuals, confounding our ability to make inferences about the timing of the underlying neural processes. Here, we show that resting-state fMRI signals contain information about HRF temporal dynamics that can be leveraged to understand and characterize variations in HRF timing across both cortical and subcortical regions. We found that the frequency spectrum of resting-state fMRI signals significantly differs between voxels with fast versus slow HRFs in human visual cortex. These spectral differences extended to subcortex as well, revealing significantly faster hemodynamic timing in the lateral geniculate nucleus of the thalamus. Ultimately, our results demonstrate that the temporal properties of the HRF impact the spectral content of resting-state fMRI signals and enable voxel-wise characterization of relative hemodynamic response timing. Furthermore, our results show that caution should be used in studies of resting-state fMRI spectral properties, because differences in fMRI frequency content can arise from purely vascular origins. This finding provides new insight into the temporal properties of fMRI signals across voxels, which is crucial for accurate fMRI analyses, and enhances the ability of fast fMRI to identify and track fast neural dynamics.

## Editor's evaluation

This manuscript addresses the important issue of hemodynamic response function (HRF) variability across brain areas and will be valuable to researchers who use fMRI and other types of functional imaging that rely on neurovascular coupling. Using simulations and experiments, the authors provide compelling evidence that differences in the HRF can impact spectrum-based metrics such as ALFF and fALFF. A better understanding of the variability of the HRF is critical for the proper interpretation of activation onset times and of differences observed in clinical populations where both neural and vascular alterations can be expected.

**eLife digest** Functional magnetic resonance imaging (fMRI) is a tool that can be used to non-invasively measure the activity of the human brain. Active parts of the brain require more oxygen, which increases blood flow to these areas. fMRI can detect these changes, and its signal reflects the coupling between brain activity and changes in blood flow.

The mechanism that couples brain activity to blood flow is known as the 'hemodynamic response', and its timing varies across the brain. Therefore, to interpret fMRI signals correctly and use them to measure underlying brain activity, it is necessary to understand how the response changes across the brain.

Current methods for probing hemodynamic response variation are either limited to specific brain regions or require patients to hold their breath – something not all groups of patients can do. To solve this problem, Bailes et al. investigated whether resting-state fMRI signals contain information about how the hemodynamic response changes across the brain. This information could then be used to better infer brain activity from fMRI measurements.

The experiments showed that resting-state fMRI signals can be used to characterize and predict the timing of the hemodynamic response. Specifically, the frequencies in resting-state fMRI signals are impacted by changes in the hemodynamic response and can therefore be used to predict hemodynamic timing. Additionally, Bailes et al. showed that these predictions are better than those obtained in experiments requiring patients to hold their breath, which is the current gold standard. The findings also demonstrate that the information from the frequencies of resting-state fMRI signals should be interpreted carefully, as differences in these frequencies can have a non-neural origin.

Bailes et al. propose a highly generalizable approach for mapping and predicting variations of the hemodynamic response across the whole brain. These findings provide insights into the time-related properties of fMRI signals that are crucial for accurate analyses. This will be of particular importance as the field moves towards fMRI studies focused on rapid neural dynamics and higher-level cognition.

## Introduction

Functional magnetic resonance imaging (fMRI) enables non-invasive measurement of human brain activity via the hemodynamic response. When activity in a population of neurons changes, these changes give rise to the blood-oxygenation-level-dependent (BOLD) signal measured in most fMRI studies (*Ogawa et al., 1990*). Thus far, however, BOLD fMRI has exhibited relatively limited ability to provide the fine-grained temporal information necessary for deepening our understanding of brain dynamics. This is due to the fact that the signals obtained from BOLD fMRI are not direct measures of neural activity, but rather reflect the coupling between neuronal activity and the hemodynamic response, which evolves on a time course of seconds (*Kwong et al., 1992*; *Ogawa et al., 1990*). This coupling between neural activity and the BOLD signal can be represented by the hemodynamic response function (HRF) (*Aguirre et al., 1998*; *Handwerker et al., 2004*). The properties of the HRF depend on many interconnected factors, including the effects of local vascular architecture and cerebrovascular dynamics, that vary substantially across the brain and between individuals (*Aguirre et al., 1998*; *Handwerker et al., 2004*; *Len and Neary, 2011*; *Logothetis et al., 2001*). The relative timing and shape of the HRF, therefore, also varies considerably across brain regions and even between neighboring voxels (*Buckner et al., 1998*; *Lee et al., 1995*; *Birn et al., 2001*; *Siero et al., 2011*; *Miezin et al., 2000*; *Siero et al., 2015*). Hemodynamic response temporal lag variation is substantially larger than many neural effects of interest, introducing variability on the order of several seconds (*Buckner et al., 1998*; *Lee et al., 1995*; *Birn et al., 2001*; *Siero et al., 2011*). Thus, to enable inferences about the relative timing of neural activity using signals obtained from BOLD fMRI, it is crucial to understand the variations in HRF timing across the brain.

Advances in acquisition technology now allow high-resolution whole brain fMRI data to be acquired at fast (<500ms) rates (*Polimeni and Lewis, 2021*; *Chen et al., 2019*; *Barth et al., 2016*; *Chiew et al., 2018*; *Hennig et al., 2007*; *Setsompop et al., 2016*), suggesting that fMRI could provide a unique tool to noninvasively track temporal sequences of neural activity (*Menon et al., 1998*) across the entire brain. Indeed, recent studies have revealed highly structured temporal dynamics using fMRI and suggest that fMRI can enable whole-brain mapping of temporal sequences (*Lee et al.,*

*2013*; *Setzer et al., 2021*; *Raut et al., 2021*; *Mitra et al., 2016*; *Vizioli et al., 2018*). Furthermore, hemodynamic signals have been shown to contain more information about fast and high-frequency activity than previously thought (*Lee et al., 2013*; *Smith et al., 2012*; *Chen and Glover, 2015*; *Lewis et al., 2016*; *Sasai et al., 2021*). Studies examining individual brain regions have demonstrated that fMRI can achieve impressive temporal precision within regions, on the order of 100ms (*Lin et al., 2013*), meaning that high fidelity temporal information is present within these hemodynamic signals. However, a key remaining challenge is that the hemodynamic differences across the brain confound our ability to infer the timing of the underlying neural activity from BOLD fMRI. Specifically, if a given brain region shows earlier BOLD activity, it could be due to faster neural activity or simply due to a faster hemodynamic response in that region. Fully exploiting the higher temporal resolution provided by fast fMRI techniques will therefore ultimately require accounting for differences in the temporal dynamics of the hemodynamic response across the whole brain.

Despite the well-known heterogeneity of hemodynamic timing, most common analysis approaches for BOLD fMRI data assume a standard, canonical HRF shape throughout the brain (*Glover, 1999*). This approach is understandable, since the true HRF is not known, but it nevertheless cannot account for the vascular confound introduced by hemodynamic response variability. Incorrect assumptions about the shape and timing of the HRF can lead to incorrect inferences regarding the underlying neural activity (*Gonzalez-Castillo et al., 2012*; *Lindquist et al., 2009*; *Handwerker et al., 2012*), and studies that assume a whole brain canonical HRF are unable to decouple the neural and vascular components of the BOLD signal (*Handwerker et al., 2004*; *Rangaprakash et al., 2018*; *Rangaprakash et al., 2017*; *Deshpande et al., 2010*; *Chang et al., 2008*). Even when using flexible modeling approaches, such as basis sets or finite impulse response models, it is not possible to determine whether a given region's faster fMRI response reflects fast neural activity, or simply faster local neurovascular coupling (*Handwerker et al., 2012*).

The fact that most studies do not account for variations in HRF dynamics is largely due to methodological challenges. Previous work has demonstrated that it is possible to quantify hemodynamic lags across brain regions, and even on a voxel-wise level, to detect the relative order of BOLD responses with high temporal precision (*Siero et al., 2011*; *Miezin et al., 2000*; *Siero et al., 2015*; *Lin et al., 2013*; *Chang et al., 2008*; *Sicard and Duong, 2005*; *Lin et al., 2018*; *Misaki et al., 2013*; *Kastrup et al., 1999*; *Liu et al., 2017*; *Bright et al., 2009*; *Chen et al., 2021*; *Wu et al., 2013*; *Wu et al., 2021*). One such method is the use of a stimulus paradigm that drives activity in particular brain regions where the neuronal response properties are relatively well understood and controlled such as primary sensory or motor cortices (*Handwerker et al., 2004*; *Birn et al., 2001*; *Siero et al., 2011*; *Miezin et al., 2000*; *Lin et al., 2018*). However, this approach cannot be applied to the majority of the brain, where the neuronal response properties are not known ahead of time. Alternatively, using a breath hold or similar hypercapnic challenge can modulate cerebral blood flow (CBF) to all vascularized regions with minimal changes in cerebral metabolic rate of oxygen ($CMRO_2$), allowing mapping of vascular latencies (*Chang et al., 2008*; *Sicard and Duong, 2005*; *Kastrup et al., 1999*; *Liu et al., 2017*; *Bright et al., 2009*; *Chen et al., 2021*; *Pinto et al., 2021*). However, breath hold tasks are not suitable for all subject populations, as some patients may have difficulty complying with the breath hold task. Furthermore, breath hold tasks modulate and measure cerebrovascular reactivity (CVR), which contributes to neurovascular coupling but is a distinct process. While neurovascular coupling reflects the alterations in local hemodynamics that occur in response to changes in neural activity, CVR is specifically a measure of a blood vessel's capacity to dilate and constrict in response to a vasoactive stimulus, and does not include the extensive metabolic and molecular factors that also drive neurovascular coupling (*Pinto et al., 2021*; *D'Esposito et al., 2003*; *Iadecola, 2017*). In fact, there is evidence that while CVR is affected by healthy aging, some metrics of neurovascular coupling are not, hinting that distinct mechanisms may shape these two patterns (*Stefanidis et al., 2019*; *West et al., 2019*; *Grinband et al., 2017*).

A task-free, neurovascular-based approach for detecting the lags of intrinsic neurovascular coupling would therefore be broadly relevant for analyzing fMRI data. A potential alternative route towards identifying local hemodynamic properties is to examine the properties of resting-state fMRI data. Resting-state fMRI signals reflect neurovascular coupling induced by spontaneous neural activity (*Mateo et al., 2017*; *Ma et al., 2016*) and confer the additional benefit of being task-independent, which makes it a viable scan type for patient populations. Moreover, unlike stimulus- or task-based

paradigms for mapping local hemodynamic response timings in specific brain regions, resting-state fMRI can be used to examine the HRF across the whole brain. These advantages have prompted past research into the utility of resting-state fMRI signals to estimate the HRF itself with prior work utilizing deconvolution approaches to explore HRF timing in resting-state data (*Wu et al., 2013*; *Wu et al., 2021*; *Sreenivasan et al., 2015*). However, these approaches require assumptions about the underlying neural events, which are not known. We therefore investigated whether intrinsic signatures of local neurovascular coupling dynamics are present in the resting-state fMRI signal.

Our goal was to understand whether information about the temporal dynamics of the hemodynamic response is present in resting-state fMRI data. We first used simulations of the BOLD response to illustrate how distinct, physiologically relevant HRF shapes should produce marked differences in the frequency content of resting-state signals. Next, we verified this result in fast fMRI data collected at 7 Tesla (T) using visual stimulation to induce a neural response with known timing in primary visual cortex (V1). We quantified the temporal delay of voxels in V1 in response to a controlled, oscillating visual stimulus, and found that voxels with fast and slow hemodynamic responses exhibited distinct resting-state spectral features. We further extended our analyses to the visual thalamus (lateral geniculate nucleus, LGN) and found that this principle generalized to subcortex. To understand the potential of this information as a tool to predict the temporal properties of individual voxels, we then trained classifiers to use information from the resting-state spectrum to classify voxels as being fast or slow cortical voxels, or even faster LGN voxels. We found that resting-state signals were better predictors of voxel-wise differences in relative hemodynamic timing than latencies measured from a gold standard breath hold task. Our results establish that information about hemodynamic timing can be extracted from the frequency spectrum of resting-state fMRI signals. This demonstrates that resting-state fMRI can provide a way to understand and predict the temporal dynamics of the HRF across the brain, which is critical for interpreting neural activity using BOLD fMRI.

## Results

### The temporal dynamics of the HRF profoundly impact the spectrum of simulated BOLD responses

Previous modeling work has illustrated that narrower HRFs should result in BOLD responses containing more high frequency power (*Chen and Glover, 2015*), suggesting that local variations in HRF timing should manifest as local variations in the frequency content of BOLD signals. We therefore hypothesized that the frequency spectrum of BOLD dynamics in the resting-state can be used to infer the relative timing of the task-driven hemodynamic response. We first aimed to illustrate this property by simulating the BOLD frequency response using HRFs with different temporal dynamics. If we assume that the BOLD response is a linear time-invariant system, we can compute the BOLD response as a convolution between a given input (i.e. the stimulus) and the characteristic input response of the system (i.e. the HRF). Then, by varying the frequency of the stimulus we can construct a spectrum of the BOLD frequency response for HRFs with faster or slower dynamics (*Figure 1A*). We performed this simulation using six different HRFs (*Figure 1B*) with physiologically representative values for their time-to-peak (TTP), full width at half maximum (FWHM), and amplitude (*Siero et al., 2011*). We observed that while HRFs with faster dynamics produced less power in the low frequency bands compared to those with slower dynamics, they also showed a shallower decline in power at higher frequencies (*Figure 1C*). Furthermore, this effect was preserved when we normalized the different HRFs to have the same peak amplitude (*Figure 1—figure supplement 1A*), demonstrating that this phenomenon is not solely due to the higher amplitude of slower HRFs (*Figure 1—figure supplement 1B*). This effect was also preserved if we accounted for the 1/f-like spectral pattern that neural activity displays (*Figure 1—figure supplement 1C–E*). And finally, although prior work has shown that the TTP and FWHM of the HRF are correlated (*Siero et al., 2011*; *de Zwart et al., 2005*), we also tested the effect of holding one parameter constant (*Figure 1—figure supplement 2*). Varying the FWHM had a more profound impact on the frequency spectrum compared to varying the TTP (*Figure 1—figure supplement 2*), as expected due to the higher frequency content of narrower HRF shapes, although short TTPs also had a smaller effect on the spectrum. These simulations demonstrate that there should be profound differences in the relative power at low versus high frequencies for voxels with fast vs. slow

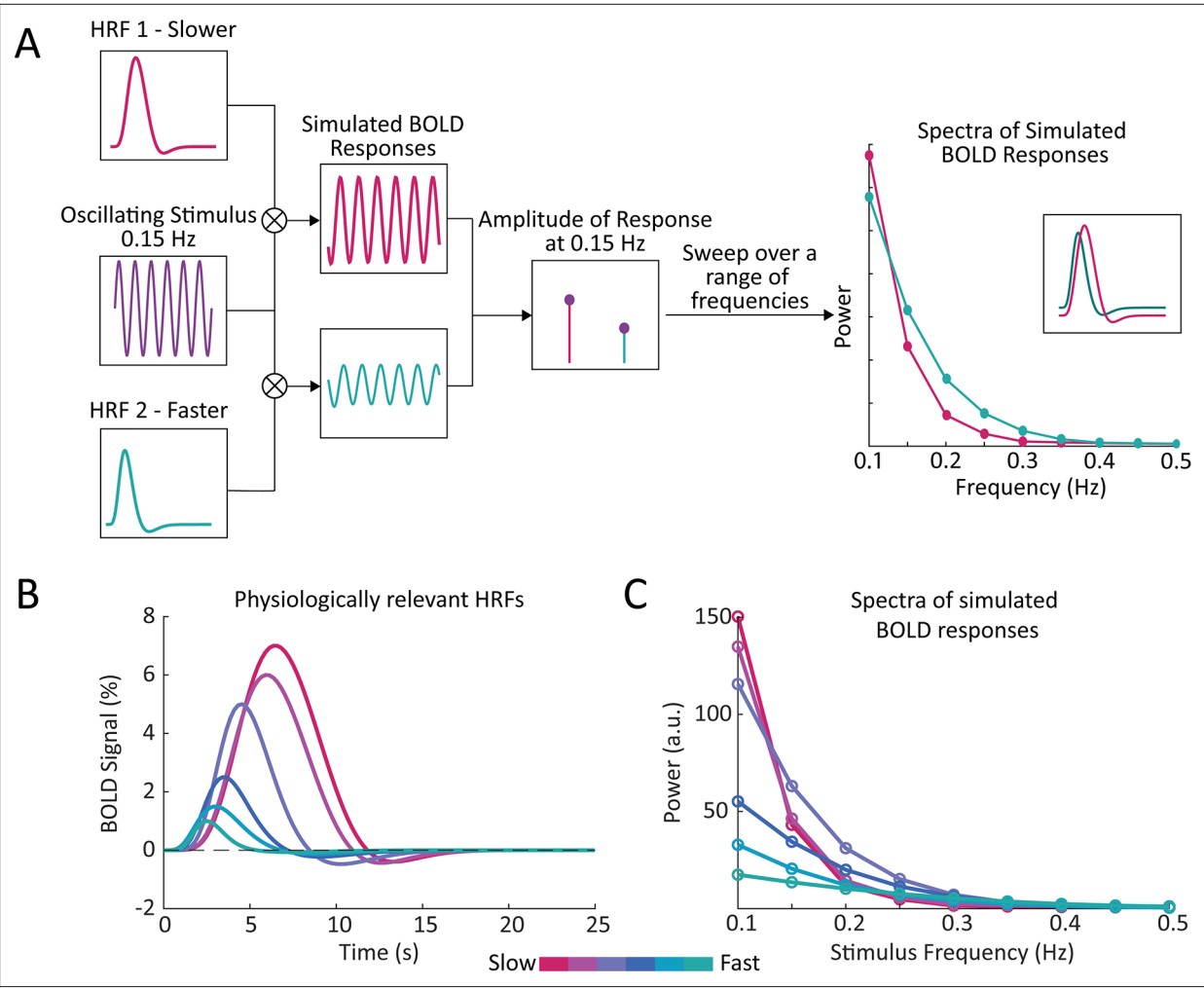

**Figure 1.** Simulations show that the temporal properties of the hemodynamic response function affect the frequency spectrum of the BOLD signal. (**A**) We generated a simulated BOLD response to determine the response amplitude of each HRF to each neural frequency. By convolving a given HRF with an oscillating stimulus, and sweeping across a range of frequencies, we generated a frequency spectrum of the simulated BOLD responses. This simulation was repeated using HRFs with varying temporal properties, to compare the frequency spectrum of simulated BOLD responses with faster or slower hemodynamic responses. (**B**) We generated a range of HRFs with physiologically plausible timings and amplitudes (*Siero et al., 2011*). (**C**) We found that temporal properties of the HRF had noticeable effects on the simulated spectra, particularly under 0.2 Hz.

The online version of this article includes the following figure supplement(s) for figure 1:

**Figure supplement 1.** Simulation results are robust to changes in HRF amplitude and 1 /f decay of stimulus amplitude.

**Figure supplement 2.** Impact of varying either TTP or FWHM on the power spectrum of simulated BOLD signals.

HRFs, which we hypothesized can be quantified and related to the temporal dynamics of the hemo-dynamic response.

## Features of the resting-state spectrum show significant differences between fast and slow voxels

Based on these simulation results, we then examined the frequency content of resting-state fMRI data. Spontaneous BOLD fluctuations captured in resting-state fMRI are linked to fluctuations in neuronal activity (*Mateo et al., 2017*; *Ma et al., 2016*; *Fox and Raichle, 2007*), and accordingly, reflect the neurovascular coupling mechanisms that link neural fluctuations to BOLD fluctuations. To empirically test the prediction that fast and slow voxels should have distinct frequency content in the resting-state, we first used a task paradigm to identify voxels in V1 with consistently fast or slow hemodynamic responses. To drive continuous oscillations in V1 we presented the subjects with a 12 Hz counterphase

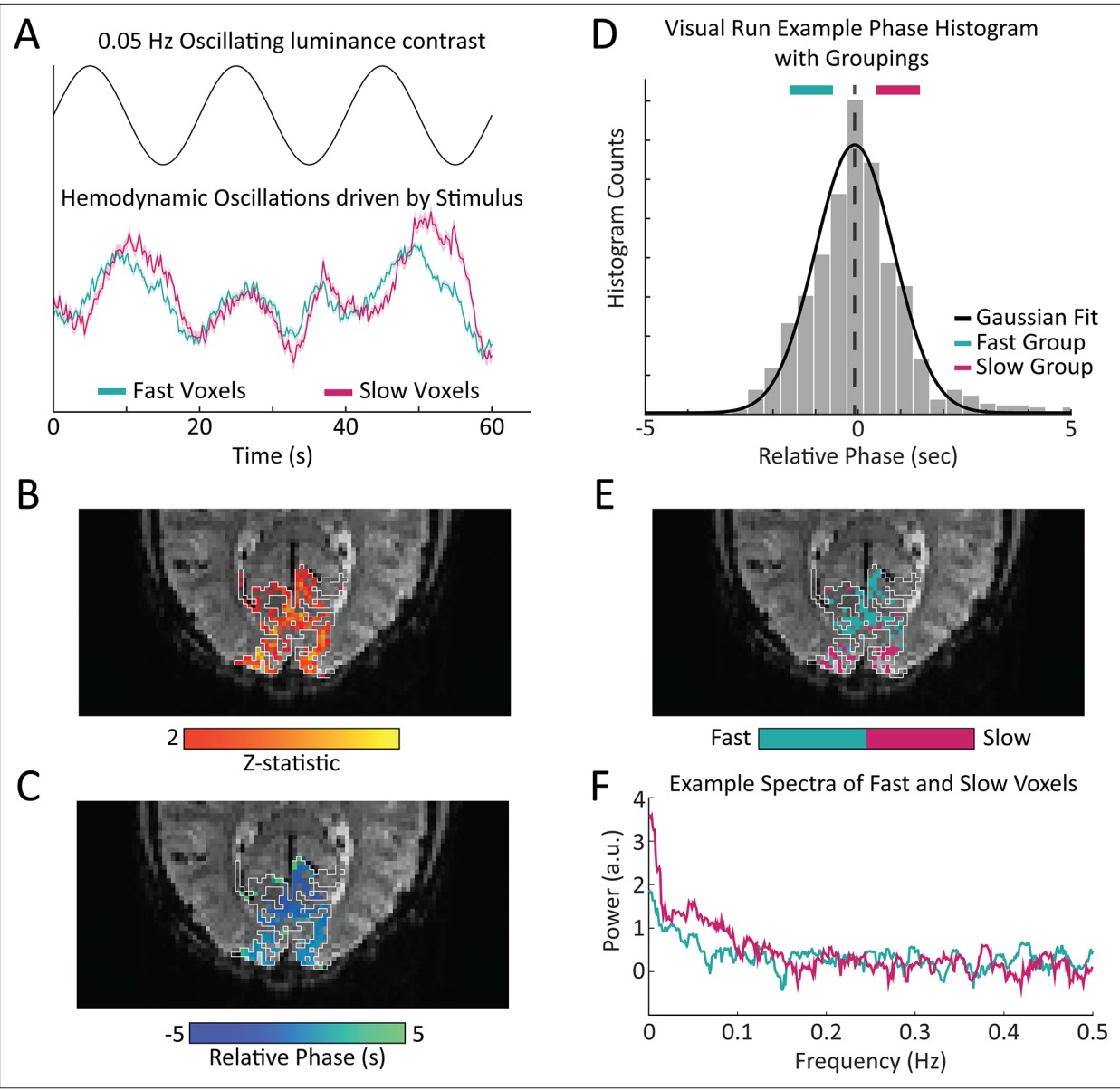

**Figure 2.** Experimental design: oscillating visual stimuli identify fast- and slow-responding voxels in V1. (**A**) Subjects viewed a flickering checkerboard with oscillating luminance contrast to drive neural oscillations in V1. Some voxels showed a faster response to the visual stimulus and other showed a slower response, with a noticeable difference in the temporal dynamics of the mean response in these groups. Shading represents standard error. (**B**) Example of a functional localizer in one subject with the white lines denoting the outline of the primary visual cortex (**V1**) based on anatomical segmentation. One visual stimulus run was used as a functional localizer to identify stimulus-driven voxels in V1. (**C**) For all stimulus-driven voxels in V1, the phase of the response to the visual stimulus was calculated from the average of the visual stimulus runs not used as the functional localizer, corresponding to the local hemodynamic delay. (**D**) We defined groups of 'fast' and 'slow' voxels using a Gaussian fit to the histogram of phases. Histogram shows example from one representative subject. (**E**) Example map of fast and slow voxels generated for a single subject. (**F**) Frequency spectrum of a representative slow and fast voxel's resting-state signal, showing a difference in power drop off across frequencies, with a steeper slope for the slower voxel.

flickering radial checkerboard with the luminance contrast of the checkerboard modulated in time as a sine wave of 0.05 Hz (*Figure 2A*, top). We used a combination of an anatomical and functional localizer to identify stimulus-driven voxels in V1 (*Figure 2B*). For those voxels that were significantly driven by the stimulus, we then calculated the phase lag, or relative response latency to the stimulus (*Figure 2C*). Consistent with prior studies (*Birn et al., 2001*; *Lewis et al., 2016*; *de Zwart et al., 2005*) we found a wide range of hemodynamic response lags within V1 (*Figure 2A*, bottom). From this distribution, we extracted groups of fast and slow responding voxels (*Figure 2D–E*). Then, for each

voxel identified as fast or slow within the task run, we calculated that voxel's frequency spectrum in the resting-state. *Figure 2F* shows the resting-state spectrum of a representative fast and slow voxel from a single subject where the differences in the spectra, particularly the distinct slope under 0.2 Hz, are visible.

Our simulations had predicted a difference in the overall frequency content of resting-state fMRI signals in voxels with fast versus slow HRFs. To quantify this property across voxels, we sought to generate a set of spectral features that could capture these resting-state spectral dynamics. We constructed four spectral features to capture spectral properties: the slope using a linear fit under 0.2 Hz, the exponent of an aperiodic 1 /f fit, the amplitude of low frequency fluctuations (0.01–0.1 Hz power; ALFF) (*Zang et al., 2007*), and the fractional ALFF (ratio of 0.01–0.08 Hz to 0–0.25 Hz; fALFF) (*Zou et al., 2008*; *Figure 3*). Each feature of the resting-state spectra revealed significant differences (Wilcoxon rank-sum test, $p<0.05$) between fast and slow voxels across subjects. The slope showed significant differences between the fast and slow voxels within each individual subject (15/15), while the aperiodic exponent, ALFF, and fALFF showed significant differences in 14, 11, and 13 subjects, respectively. (See *Supplementary file 1* for p-values). Each of these features reflects information about the frequency content at high and low frequencies, suggesting that this was an effective metric for differentiating voxels with fast or slow hemodynamic responses. Notably, the most effective features for distinguishing fast and slow voxels were the ones that explicitly captured the relative difference in high-frequency vs. low-frequency power.

## Faster hemodynamic responses in thalamus are also reflected in shallower frequency spectra

A key strength of fMRI is its ability to noninvasively image activity in the subcortex. Having established that the resting-state spectrum contained signatures of local hemodynamic response timing within V1, we next aimed to test whether this principle extended to the visual thalamus, specifically the LGN. Prior studies have shown that LGN has faster hemodynamic responses than V1 (*Lau et al., 2011*; *Yen et al., 2011*; *Lewis et al., 2018*); however, due to its small size and lower signal-to-noise ratio, extracting its spectral features accurately could be more challenging. We generated individual masks of the LGN within each subject using the individual anatomical segmentation (*Fischl, 2012*) and a functional localizer (*Figure 4A*). To confirm the presence of stimulus-locked oscillations in the LGN and to assess the relative timing of its response, we first examined the latency of the average response to the visual stimulus in the LGN, as compared to the fast and slow groups in V1. We observed that the LGN peaked before both the fast and slow groups in V1 (*Figure 4B*), consistent with prior work demonstrating faster hemodynamics in thalamus (*Lau et al., 2011*; *Yen et al., 2011*; *Lewis et al., 2018*). Then, to test whether this faster hemodynamic response was similarly linked to flatter resting-state spectra, we compared the LGN voxels' resting-state features to the previously identified fast and slow voxels in the cortex. We found that for all subjects there were clear differences in each spectral feature within the LGN compared to both the fast and slow voxels of the visual cortex (*Figure 4C–F*; See *Supplementary file 1* for p-values). The feature that had the poorest sensitivity to differences between the LGN features and cortical features is ALFF, which could be explained by ALFF's higher sensitivity to non-neural noise sources (*Zou et al., 2008*). We thus observed even shallower frequency slopes for the fast LGN voxels – again consistent with our simulation results, demonstrating that this pattern held not just within V1 but even extended to the LGN of the thalamus. This observation was also robust to controlling for the higher thermal noise in LGN signals (*Figure 4—figure supplement 1*).

## Resting-state spectral information better characterizes neurovascular coupling delays than a breath hold task

Perhaps the most established method of mapping hemodynamic latencies across the brain is using a breath hold task to quantify cerebrovascular reactivity (*Chang et al., 2008*; *Kastrup et al., 1999*; *Liu et al., 2017*; *Bright et al., 2009*; *Chen et al., 2021*; *Pinto et al., 2021*). Therefore, we tested whether similar information about hemodynamic latencies found using the resting-state spectra could be found in data from the breath hold task. To determine this, we first mapped the vascular latency in response to the breath hold task on a voxel-wise basis across the brain (*Chang et al., 2008*). We then compared each voxel's relative vascular latency from the breath hold task across the three groups

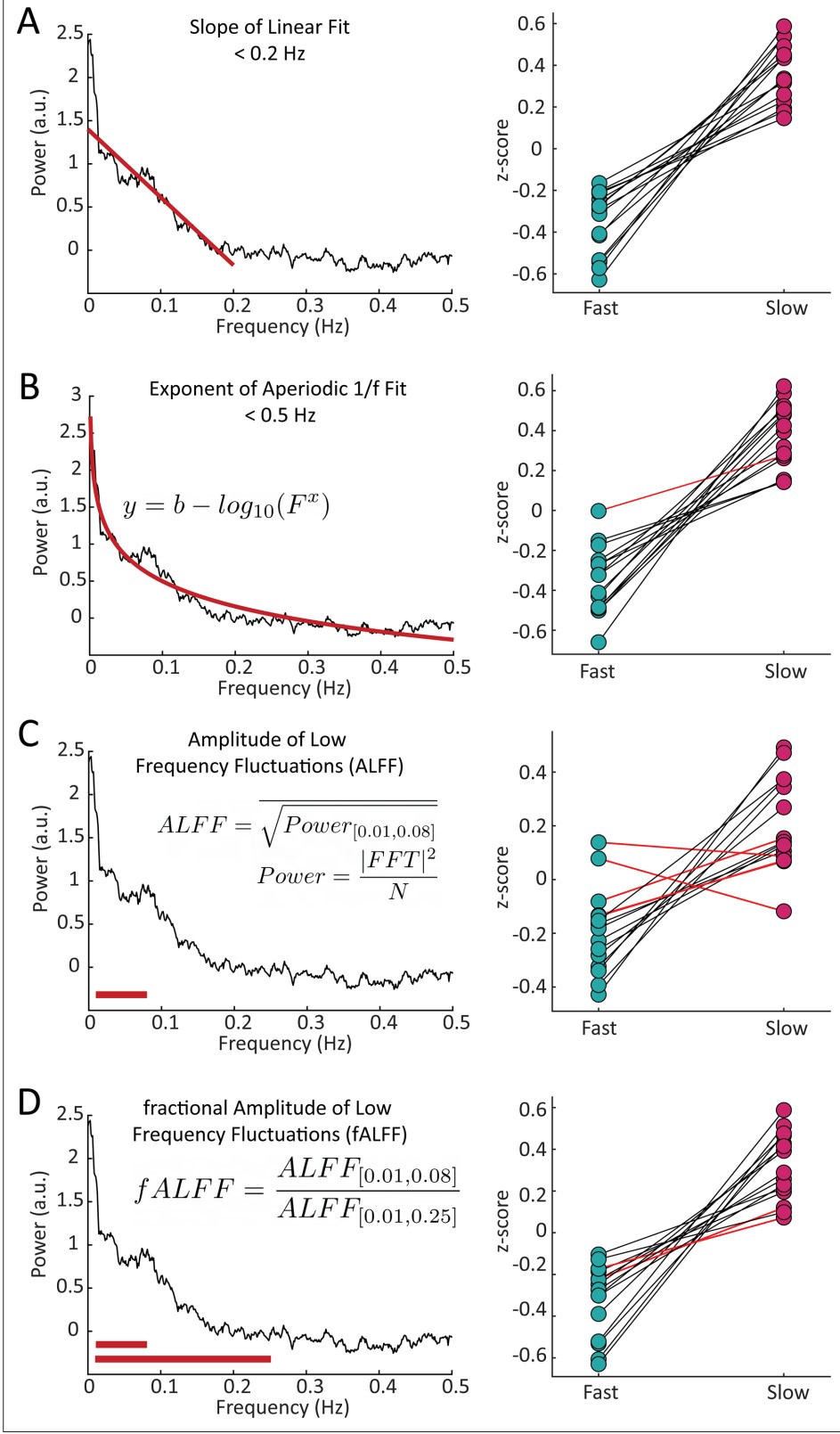

**Figure 3.** Features of the resting-state spectrum differed between fast and slow voxels in each subject. For each subject, we calculated four features of the resting-state frequency spectrum and compared the values between the task-defined fast and slow voxels using a Wilcoxon rank sum test. For the (**A**) slope, using a linear fit of frequency spectrum under 0.2 Hz, 15/15 subjects showed significant differences; (**B**) exponent of an aperiodic

*Figure 3 continued*

1 /f fit under 0.5 Hz, 14/15 subjects showed significant differences; (**C**) amplitude of low frequency fluctuations (ALFF), 11/15 subjects showed significant differences; and (**D**) fractional ALFF, 13/15 subjects showed significant differences. Black lines indicate a significant difference in a given subject (Wilcoxon rank-sum test, p<0.05) and red lines indicate a non-significant difference.

The online version of this article includes the following figure supplement(s) for figure 3:

**Figure supplement 1.** The same pattern of frequency content for fast and slow voxels replicated in an independent dataset acquired at 3T.

**Figure supplement 2.** There is no consistent pattern of relationship between fast and slow voxels' estimated noise floor of the frequency spectrum.

**Figure supplement 3.** Example subjects showing significant (p<0.05) correlations between each spectral feature and phase on a voxel-wise basis.

identified in the visual task – the fast and slow cortical voxels and LGN voxels. We found that some subjects showed the expected temporal sequence of activation, where LGN voxels and fast cortical voxels respond earlier than slow cortical voxels (*Figure 5A*), but this effect was not present in all subjects (*Figure 5B*). Even among the subjects that exhibited the expected order of activation, few of them showed individual-level significant differences in breath hold latency between the groups (*Figure 5C*). Specifically, in only 7/15 subjects was the average breath hold latency of fast cortical voxels significantly faster compared to slow cortical voxels. Furthermore, the breath hold latency in LGN voxels was slower than expected in some subjects: it was significantly slower than fast cortical voxels in 5 subjects and significantly slower than slow cortical voxels in 1 subject (Wilcoxon rank-sum

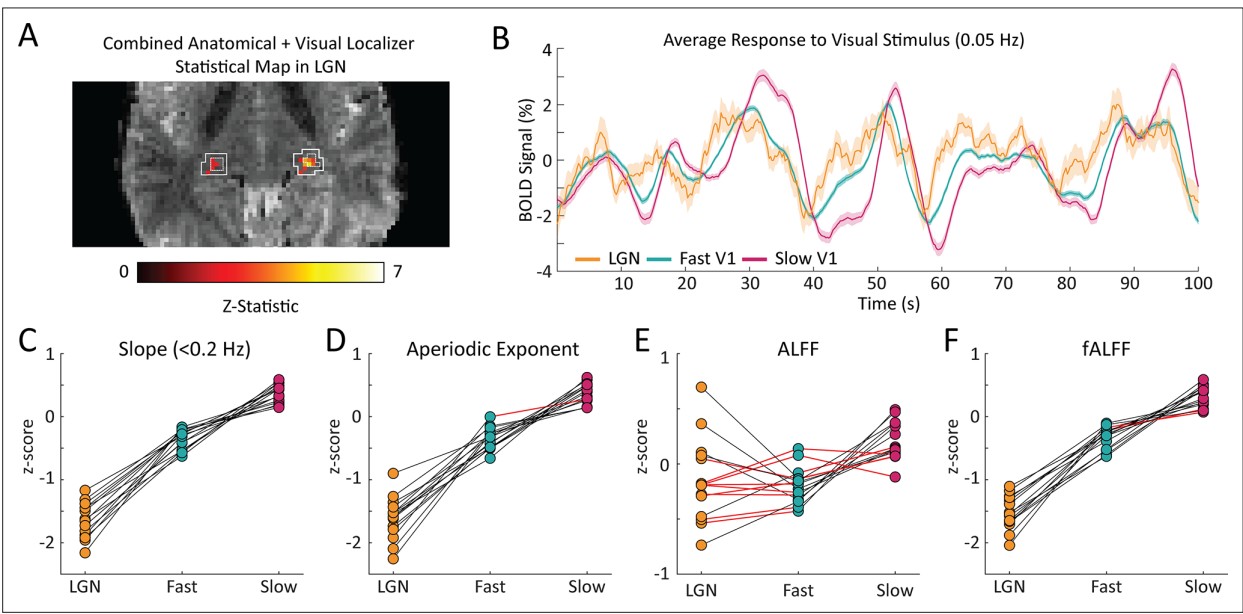

**Figure 4.** The coupling of response timing and resting-state spectral content is maintained in the LGN. (**A**) Example LGN localizer in one subject showing uncorrected z-statistic in the localizer run within the anatomical mask of LGN defined by the white outline. (**B**) Average time series of fast and slow V1 voxels compared to LGN voxels for an example subject. LGN displayed a fast visually-driven response, leading even the earliest cortical voxels. Time series are smoothed for display using a 10-point moving average. Shading represents standard error. (**C–F**) For each subject we calculated the four resting-state spectral features for the LGN and compared them to the fast and slow voxels in V1. For the (**C**) slope, 15/15 subjects showed significant differences between fast vs. LGN and slow vs. LGN; (**D**) exponent of an aperiodic 1 /f fit, 15/15 subjects showed significant differences between fast-LGN and slow-LGN; (**E**) ALFF, 5/15 subjects showed significant differences between fast vs. LGN and 7/15 between slow vs. LGN; and (**F**) fALFF, 15/15 subjects showed significant differences between fast-LGN and between slow-LGN. Black lines indicate a significant difference (Wilcoxon rank-sum test, p<0.05) and red lines indicate a non-significant difference.

The online version of this article includes the following figure supplement(s) for figure 4:

**Figure supplement 1.** Accounting for thermal noise does not significantly change the estimated slope of the frequency spectrum under 0.2 Hz in V1 or LGN voxels.

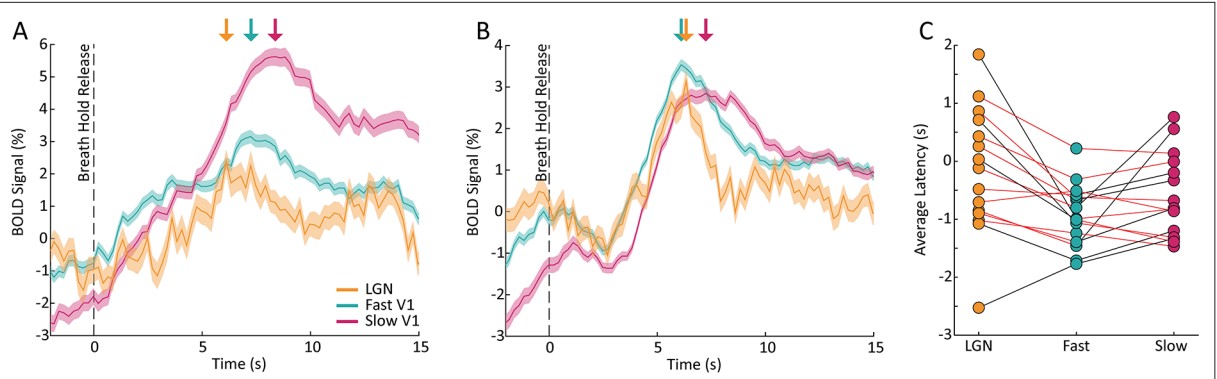

**Figure 5.** Breath hold vascular latencies yield less robust characterization of task-driven hemodynamic response lags. (**A**) Plot of one subject's average BOLD response to the breath hold task in fast (teal) and slow (pink) cortical voxels as well as LGN voxels (yellow). The shaded areas are the standard error across voxels of that group. Colored arrows denote the peak of the response to the breath hold. The LGN time series peaks slightly earlier than the fast cortical voxels and both the LGN and fast cortical voxels peak well before the slow cortical voxels. This sequence of activation matches what is expected based on the hemodynamic lags across these structures. (**B**) Plot of one subject's average BOLD response to the breath hold task where the order of activation is not as expected. While the slow cortical voxels reached their peak last, the fast cortical voxels peaked before LGN voxel, meaning that the breath hold would not accurately predict latency in this subject. (**C**) Comparison of the average breath hold latency in fast, slow, and LGN voxels in all subjects. For 9/15 subjects, the average latency of the fast cortical voxels was less than the slow cortical voxels, as expected, and 7 of these 9 had a significant difference. For only 2/15 subjects, the average latency of the LGN voxels was less than fast voxels, as expected, but only 1 of these 2 had a statistically significant difference. For 4/15 subjects, the average latency of slow cortical voxels was larger than LGN voxels, and 1 of these 4 had a statistically significant difference. (Wilcoxon rank-sum test, p<0.05).

The online version of this article includes the following figure supplement(s) for figure 5:

**Figure supplement 1.** Subject motion during the breath hold task was significantly worse than the visual stimulus task.

test, p<0.05). These results demonstrate that although the vascular latencies derived from the breath hold task do show significant differences between fast and slow cortical voxels and LGN voxels across the group, this effect is less robust in individual subjects than the resting-state spectral features. Additionally, not all subjects demonstrated the expected order of latencies between the three groups – with LGN first followed by fast cortical and, lastly, slow cortical voxels. Taken together, these results suggest that the features of the resting-state spectrum capture additional information about local differences in neurovascular coupling delays.

## Features of the resting-state spectrum can predict voxels with fast or slow hemodynamic response timing

Our results showed consistent signatures of hemodynamic response latency in the resting-state fMRI signal, suggesting that this information could potentially be used to predict local neurovascular latencies. Indeed, we found that these spectral features were significantly correlated with the absolute timing of their hemodynamic responses (*Figure 3—figure supplement 3*, *Supplementary file 3*). To investigate the utility of the resting-state spectrum to predict the temporal dynamics of the HRF, we tested whether support vector machines (SVMs) could classify slow, fast, and fastest (LGN) voxels using information from the resting-state spectrum. First, we trained a SVM to classify slow, fast, and LGN voxels based on the four features of the resting-state spectra identified in *Figure 3*. We found that the classifier validation accuracies, both within each individual subject and on the dataset that combined all subjects, were well above chance (*Figure 6*), demonstrating robust prediction of local hemodynamic delays (per-subject accuracy in *Supplementary file 2*). We next considered whether correlated information between neighboring voxels could be contributing this prediction, to test whether we could generalize to distant voxels. To control for instances of voxels in the test set being in close proximity to voxels in the training set, we trained a new set of models using voxels in a single hemisphere (left or right) as the training set, and voxels in the other hemisphere as the testing set. Following this procedure, the number of voxels for training and testing was often closer to a 50–50 split than an 80–20 split; however, validation accuracies both within individual subjects and across subjects nevertheless remained well above chance. Interestingly, the overall performance across

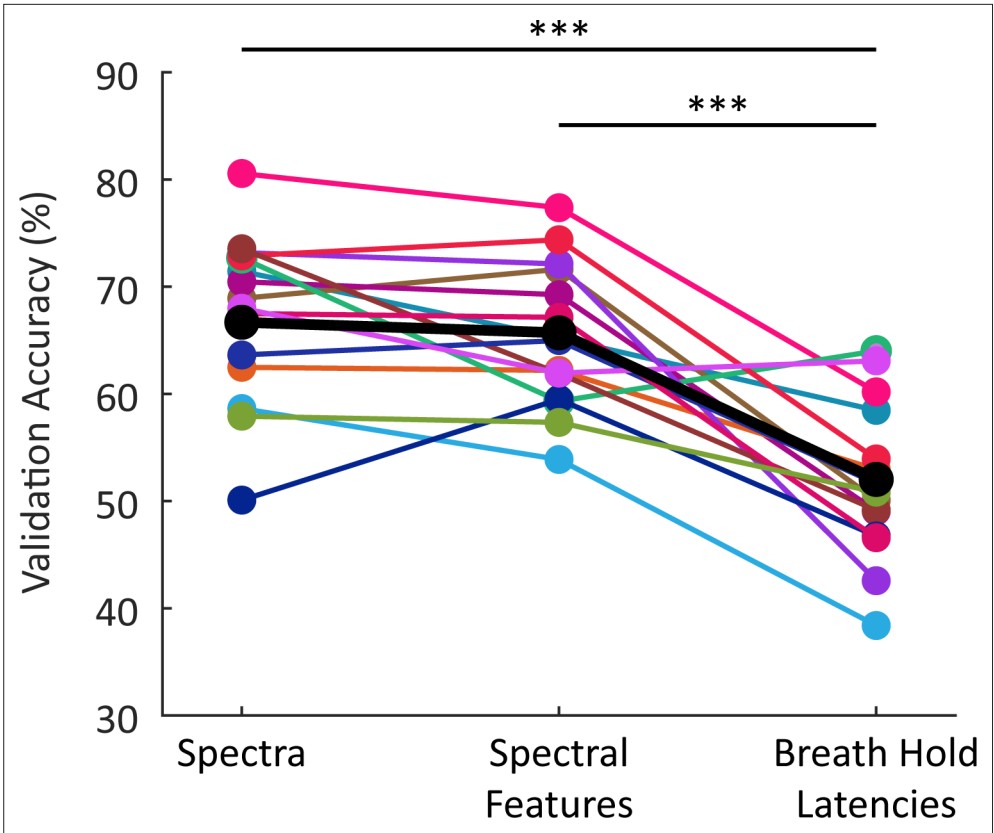

**Figure 6.** Average classification accuracy of SVM classifier trained using different features. The classification accuracies reported are the average validation accuracy across 1000 bootstraps. The black markers indicate the models trained using resting-state spectral features from all subjects combined into one model; the colored markers show results from individual subjects (n=15). Both models trained using features of the resting-state spectrum performed significantly better than the model trained using the breath hold latencies; however, there was no significant difference in performance between the two models trained using information from the resting-state spectrum (=0.283). All classifiers perform significantly above chance (33%). *** p<0.0005 (Wilcoxon rank-sum test).

The online version of this article includes the following figure supplement(s) for figure 6:

**Figure supplement 1.** Classification accuracies of SVM classifier when requiring that training and test voxels be physically separated.

subjects was higher, suggesting better generalization across subjects when training on spatially separated voxels (*Figure 6—figure supplement 1*).

Regression models aiming to predict specific response timing (rather than classifying fast vs. slow) were significantly above chance but still performed poorly, with an average coefficient of determination of 0.104 and an average root mean squared error of 0.923 (*Supplementary file 4*), likely due to the fact that absolute HRF timing depends on task state (*Chen and Glover, 2015*). We next investigated whether our chosen features were sufficient for classification, or whether additional useful information was present in the full resting-state spectrum, by training a second SVM classifier to use each voxel's resting-state spectrum as the predictor for the classifier. The input to the classifier was the resting-state spectrum limited to up to 0.5 Hz to reduce the number of features fed into the model to avoid overfitting. Once again, we found that the accuracies, both within individual subjects and on the dataset that combined all subjects, were well above chance (*Figure 6*; per-subject accuracy in *Supplementary file 2*). The SVM classifiers trained using the spectrum trended toward slightly higher classification accuracies than the pre-selected features (*Figure 6*), but the performance of the two models was not significantly different (p=0.2828; Wilcoxon rank-sum test). We therefore concluded that our constructed spectral features are capturing key aspects of the resting-state spectrum that are important for predicting hemodynamic latency.

We next tested whether the resting-state signals performed better than the breath hold task at differentiating between fast, slow, and LGN voxels. We trained another SVM classifier to use the breath hold latencies as predictors and found that this model performed significantly above chance (*Figure 6*), consistent with prior work demonstrating its utility in predicting delays. However, it nevertheless performed significantly worse than both resting-state models (Wilcoxon rank-sum test, $p<0.0005$; per-subject accuracy in *Supplementary file 2*). This result further supported the conclusion that resting-state spectral information performs well at capturing differences in hemodynamic response timings.

## Discussion

We conclude that the frequency spectrum of resting-state fMRI signals contains rich information about local hemodynamic timing. Our simulations first illustrated the rationale for this effect: HRFs with faster temporal dynamics produce less power in the low frequency bands and a shallower attenuation of power at high frequencies. This effect is preserved even when we account for the higher amplitude of slower HRFs and the $1/f$ decay of the amplitude of spontaneous neural activity. Following this observation, we constructed several quantitative features based on the resting-state spectrum, each capturing a similar property: the relative amplitude of high vs. low frequencies, and when used as predictors for a SVM were able to classify individual voxels as fast or slow. Our findings demonstrate that the temporal properties of the HRF affect the spectral features of resting-state fMRI signals and present a framework for characterizing the temporal properties of the hemodynamic response across voxels, which is crucial for accurate fMRI analyses.

The work presented here has broad implications for fMRI studies using the frequency content of the BOLD signal to make inferences about intrinsic brain activity. Prior work has identified changes in the spectral content of fMRI signals in certain clinical populations, including major depressive disorder (*Wang et al., 2016*), mild cognitive impairment (*Han et al., 2011*), and Alzheimer's disease (*Yang et al., 2020*), and interpreted these differences as changes in intrinsic brain activity. However, as seen here, differences in hemodynamic response timings will also alter the frequency content of fMRI signals. Furthermore, prior work has shown that intrinsic timescales of BOLD activity vary across brain regions and can be used to predict individual subject patterns (*Fallon et al., 2020*). Our work suggests hemodynamic differences may contribute to these observations. Therefore, changes in the spectral content of fMRI signals can arise not just from differences in intrinsic brain activity but could also be indicators of different hemodynamic response timing.

A common theme across the spectral features we constructed was that they were sensitive to the relative contributions of low- and high-frequency power. Two features we selected (the slope of the spectrum and the exponent of the aperiodic $1/f$ fit) are direct measures of the attenuation in power towards higher frequencies. Similar information is contained in the commonly used metrics ALFF and fALFF (*Zou et al., 2008*): ALFF is a marker of the power in low frequency bands and fALFF is a measure of the ratio of high to low frequency power. While each of these features showed significant differences between fast and slow cortical voxels, only the slope and fALFF exhibited significant differences within each individual subject. These features explicitly capture the relative difference in high-frequency vs. low-frequency power, corresponding to the main prediction of our simulations. Conversely, ALFF had the poorest sensitivity within subjects which could be due to the fact that ALFF is the only feature that does not include information from both low and high frequency ranges and is instead limited to a relatively narrow band of frequencies, 0.01–0.08 Hz. Overall, our results suggest that while some information is contained in the magnitude of the fMRI signal alone, capturing the relative power in high versus low frequencies is the most important metric for predicting hemodynamic timing. Moreover, using the resting-state spectrum itself did not perform significantly better than our chosen features in classifying fast and slow voxels, suggesting that the relative power is indeed the primary contributor when predicting variations in local HRF timings.

Our results highlight the substantial variability in hemodynamic timing even within a single cortical area, and provide a way to predict this speed, but the specific reasons why individual voxels are fast or slow are not yet clear. A multitude of neurovascular and anatomical factors contribute to the variability in hemodynamic response timing across the brain. For example, the anatomical organization of the vascular network inherently introduces temporal dispersion to the hemodynamic response. As blood travels through the vascular network in response to changes in neuronal activity it not only encounters

a range of vessel diameters, but also many branch points which each introduce delays. Prior work has shown that the timing and duration of the hemodynamic response both increase towards the cortical surface while the fastest, narrowest hemodynamic responses occurred in deeper gray matter (*Siero et al., 2011*; *de Zwart et al., 2005*; *Duvernoy et al., 1981*). This suggests the possibility that the fast responding voxels identified in our study are more concentrated in deeper gray matter, although the spatial resolution of our images was not sufficient to separate voxels by cortical depth in this study. Vascular elasticity and reactivity also change across the vascular network and vary between regions further contributing to the variable timing of the hemodynamic response across the brain (*Uludağ and Blinder, 2018*; *Drew, 2019*). Arteries are surrounded by smooth muscle cells that rapidly relax and expand their diameter in response to neural activity while veins dilate much more slowly (*Drew, 2019*). Local differences in the relative concentrations of arteries and veins could therefore also introduce differences in the temporal dynamics of the BOLD signal (*Siero et al., 2011*; *de Zwart et al., 2005*; *Taylor et al., 2018*), suggesting that the slow responding voxels may be more likely to contain veins. These relationships between vascular anatomy and timing further highlight that fMRI studies analyzing spectral power or temporal autocorrelation should take into consideration how local vascular anatomy affects these metrics.

A unique advantage of fMRI is its ability to image throughout the whole brain, and our results suggest the potential for extracting more information from fMRI studies of subcortex. Due to significant improvements in both the sensitivity and spatial resolution of fMRI, an increasing number of studies are utilizing fMRI to study small, deep brain structures such as the thalamus and brainstem (*Beissner et al., 2014*; *Sclocco et al., 2018*; *Saranathan et al., 2021*; *Beissner and Baudrexel, 2014*). The ability to image these deeper brain structures in humans opens the door to studying diverse aspects of cognition associated with these deeper brain regions (*Beissner et al., 2014*; *Sclocco et al., 2018*; *Saranathan et al., 2021*; *Beissner and Baudrexel, 2014*; *Sherman, 2007*). However, these deep brain structures also have unique physiological and anatomical properties that alter their vascular dynamics (*Lau et al., 2011*; *Yen et al., 2011*; *Lewis et al., 2018*; *Duvernoy, 2009*; *Devonshire et al., 2012*). Faster hemodynamic responses are frequently reported in subcortex, but hemodynamic responses in these regions are less well characterized than in the cortex. Our analyses found that variations in HRF timing are reflected in the frequency spectra of thalamic voxels as well, despite their differing signal-to-noise ratio. This result demonstrates the utility of our approach in subcortex and could benefit neuroimaging studies of structures such as the thalamus, a target of increasing interest in fMRI.

Improvements in the temporal resolution of BOLD fMRI have also sparked interest in detecting neural sequences at sub-second timescales, which are highly relevant for many studies of cognition. Recent studies have leveraged fast fMRI to detect rapid sequences of neural events related to visual sequence detection (*Wittkuhn and Schuck, 2021*), auditory dynamics (*Frühholz et al., 2020*), and changes in arousal state (*Setzer et al., 2021*). As we continue to identify these rapid neural sequences, it will become even more crucial to consider how the hemodynamic response varies across regions to determine whether a given sequence represents regional differences in neuronal or in hemodynamic timing. Considering spectral signatures can support inference of precise timing of neural activity by providing information about relative hemodynamic latencies between voxels and regions.

We found that the resting-state signals predicted voxel-wise differences in relative hemodynamic response timing significantly better using a breath hold to estimate vascular delays, the current gold standard. This observation could be explained by multiple factors. One potential factor is subject motion (*Figure 5—figure supplement 1*). There was significantly more motion during the breath hold task compared to the visual stimulus (Wilcoxon rank-sum test, p=0.01), which could lead to breath tasks providing less reliable information. Additionally, the models trained using the breath hold latencies were only able to leverage one piece of information: the measure of vascular latency derived from cross-correlation. The information from taking the cross-correlation in the time domain may also be more sensitive to noise, especially at the fast sampling rate of our scans. However, a more important factor may be the biological mechanisms generating each signal. The breath hold task directly modulates cerebrovascular reactivity with minimal accompanying changes in $CMRO_2$, allowing for assessment of local cerebral vascular reactivity uncoupled from neuronal activation (*Kastrup et al., 1999*). CVR is important to assess in many clinical applications, and the breath hold-based approach remains a gold standard for CVR mapping (*Kastrup et al., 1999*; *Liu et al., 2017*; *Bright et al., 2009*; *Pinto*

*et al., 2021*). However, while CVR is a significant modulator of the hemodynamic response, it is only one component of neurovascular coupling, and there are many other factors and signaling pathways that also contribute. In particular, local metabolic factors and feedback mechanisms also modulate of blood flow in the brain and are not replicated in the breath hold task (*Iadecola, 2017*; *Hosford and Gourine, 2019*). The hemodynamic responses induced by neural activity may, therefore, not be identical in timing to those induced by a purely vascular signal. By contrast, the signals obtained from resting-state fMRI are coupled to underlying neural activity in an analogous manner as during a task condition (*Mateo et al., 2017*; *Ma et al., 2016*; *Fox and Raichle, 2007*). The improved performance of the resting-state-based prediction may therefore reflect that it is intrinsically more similar to a task than the breath hold condition, as the underlying biological origins of resting-state signals share common mechanisms with task-induced neurovascular coupling. Future applications may therefore benefit from continuing to use breath tasks as a gold standard to assess CVR, whereas resting-state analyses may be a better metric of neurovascular coupling.

One limitation of our study is that our approach has thus far only been validated in the visual system, which is the basis for the majority of our knowledge about the HRF. Given that the HRF has been shown to vary across the brain, the generalizability of our approach to different brain regions has yet to be established. However, studies that have empirically derived the HRF in other brain regions, such as the motor cortex (*Siero et al., 2011*) or somatosensory cortex (*de Zwart et al., 2005*), have produced HRFs with similar shapes and parameters to the double-gamma HRF utilized in our simulations (*Handwerker et al., 2004*; *Siero et al., 2011*; *de Zwart et al., 2005*; *Taylor et al., 2018*) providing evidence that this approach may be generalizable across the brain. Still, future work could investigate this approach in other primary sensory systems such as the auditory system where there are reliable stimulation paradigms. Additionally, this study used data collected at an ultra-high magnetic field strength (7T) which affords a higher signal-to-noise ratio (SNR) compared to more traditional 3T fMRI. However, we also performed the same spectral analysis in an independent dataset acquired at 3T and robustly replicated our result, demonstrating that this pattern is preserved despite a lower SNR and different acquisition parameters (*Figure 3—figure supplement 1*).

The work discussed here suggests a wide range of neuroscience applications for our approach to measuring hemodynamic timing. One logical next step would be to use the ability to characterize temporal variation in the HRF to not just predict, but to correct for vascular delays. Previous work demonstrated that correcting for varying hemodynamic latencies across the brain can affect functional connectivity analyses (*Rangaprakash et al., 2018*; *Chang et al., 2008*). Our results could further enhance removal of non-neural latency differences that confound functional connectivity metrics, both static and dynamic, and can increase confidence that the networks we are analyzing are derived from neuronal dynamics. This has become increasingly of interest as more studies use functional connectivity, particularly in resting-state, to study dynamics underlying diseases such as PTSD (*Jin et al., 2017*; *Rabinak et al., 2011*; *Maron-Katz et al., 2020*; *Zhu et al., 2017*), Alzheimer's Disease (*Greicius et al., 2004*; *Bai et al., 2008*; *Agosta et al., 2012*; *Koch et al., 2012*), Parkinson's Disease (*Helmich et al., 2010*; *Hacker et al., 2012*; *Agosta et al., 2014*; *Baggio et al., 2015*; *Putcha et al., 2015*), and others (*Filippi et al., 2019*). Since changes in neurovascular function have been observed in many disorders (*de la Torre and Mecocci, 2018*; *Gutteridge et al., 2020*; *Burrage et al., 2018*), analyzing spectral dynamics may help interpret functional connectivity differences in clinical populations.

Although we focused on the utility of resting-state spectral information to classify voxels as having relatively fast or slow hemodynamic response timings, we did also examine the correlation with absolute timing measures. We found that most subjects had significant positive correlations between each of the spectral features and the task response latency. Despite this fact, when estimating continuous predictions of absolute hemodynamic response lag, prediction performance was poor. Importantly, previous evidence has shown that the absolute timing of the HRF varies between task and resting-state conditions (*Chen and Glover, 2015*), suggesting that caution is needed in using absolute measures of hemodynamic latencies, as these may not generalize across conditions. Measuring relative differences in HRF timing might therefore be a more effective method to correct for variations in the hemodynamic response across brain regions, since task state can modulate the absolute timing of the hemodynamic response. Furthermore, the relative differences provide the key information necessary to interpret sequences of activity across brain regions, enabling examination of whether purely vascular differences are present across those regions.

Together, our results demonstrate that the resting-state fMRI signal contains information about local hemodynamic response speeds. This approach can help understand brain-wide variations in HRF dynamics, which is critical as the field moves toward a new era of fMRI studies utilizing fast fMRI to study rapid neuronal dynamics and higher level cognition.

## Materials and methods

### Simulations

Spectra of simulated BOLD responses were generated by convolving a given HRF with oscillating stimuli, ranging from 0.1 to 0.5 Hz, and taking the magnitude of the simulated BOLD response as the power at that frequency (*Figure 1A*). We used six HRFs with varying time-to-peak (TTP), full width at half maximum (FWHM), and peak percent signal changes (PSCs) to represent a range of physiologically relevant HRFs (*Figure 1B*). These properties were drawn from previous work characterizing varying HRF temporal dynamics at different cortical depths (*Siero et al., 2011*) and the values used are reported below in *Supplementary file 5*. We also normalized these HRFs by their maximum percent signal change and re-simulated the BOLD responses to create a new simulated spectrum for each HRF (*Figure 1C*). Additionally, we performed simulations to account for the dominant 1/f-like spectral pattern of neural activity by setting the amplitude of the oscillating stimuli to be 1/stimulus frequency ranging from 0.1 to 0.5 Hz (*Figure 1—figure supplement 1*).

### Subject population

All experimental procedures were approved by the Massachusetts General Hospital Institutional Review Board (protocol number: 2014P001068), and all subjects provided informed consent. Twenty-one participants were scanned in total; five were excluded for excessive motion and one was excluded for poor performance on the visual task suggesting they had closed their eyes. This left 15 subjects whose data was analyzed (mean age = 28 years, range = 22–42 years, 8 female).

### Experimental design

Subjects underwent a total of 7 functional scans: 3 visual stimulus, 2 breath hold, and 2 resting-state runs. All stimuli were programmed in MATLAB using Psychtoolbox (*Brainard, 1997*).

### Visual stimulus

Each visual stimulus functional run lasted 254 s, with the first 14 s showing a gray screen with the red fixation dot and the following 240 s consisting of the 12 Hz counterphase flickering radial checkerboard. To drive continuous neural oscillations in the visual cortex, the luminance contrast of the flickering checkerboard oscillated at a frequency of 0.05 Hz (except for one subject who was presented with 0.1 Hz oscillations). To assist the subjects with fixation, in the center of the visual field was a red dot that changed brightness at random intervals. Subjects were directed to press a button whenever the brightness of the red dot changed, and their average response time and response accuracy was reported at the end of each run. This allowed us to monitor participant engagement with the task. Each subject participated in 3 visual stimulus runs.

### Breath hold Task

For each breath hold run subjects performed eight repetitions of an adapted version of a previously established breath hold task (*Chang et al., 2008*): a block comprised of 27 s of free breathing, 3 cycles of paced breathing (3 s breathe out, 3 s breathe in), a 15 s breath hold, and, lastly, a 30-s period of free breathing. The total time for the breath hold task scans was 8.5 min. The instructions for breathing were projected to the subjects displaying the text 'Breathe Freely'; 'Breathe Out'; 'Breathe In'; and 'HOLD BREATH'. All but one subject participated in 2 breath hold runs; a single subject performed only 1 breath hold run. For 2 subjects, a single breath hold run was excluded from analyses for excessive motion defined here as greater than 0.5 mm average motion across the whole run. No additional runs were excluded for these subjects.

## Resting-state

Each subject participated in 2 resting-state runs where they were instructed to relax in the scanner with their eyes open and to try not to fall asleep. For some of the subjects, a fixation dot was presented to help minimize eye movements. Each resting-state scan lasted 8.5 minutes.

## MRI Data Acquisition

Subjects were scanned on a 7 Tesla Siemens MAGNETOM scanner with a custom-built 32-channel head coil. Anatomical images were acquired with 0.75 mm isotropic multiecho magnetization-prepared rapid gradient-echo (MEMPRAGE) protocol (*van der Kouwe et al., 2008*) with TR = 2.530ms, echo time (TE)=1.76ms and 3.7ms, inversion time (TI)=1100ms, echo-spacing=6.2ms, 7° flip angle, bandwidth = 651 Hz, in-plane acceleration *R*=2, FOV = 320 x 320 x 244 mm and a total scan time of 7:20 min. For functional runs, 15 oblique slices were positioned to target the calcarine sulcus to include primary visual cortex (V1) and angled to include the lateral geniculate nucleus (LGN) located in the thalamus. Functional runs were acquired as single-shot gradient-echo EPI with 2 mm isotropic resolution, TR = 227ms, TE = 24ms, echo-spacing=0.59ms, 30° flip angle, bandwidth = 2604 Hz, in-plane acceleration *R*=2, SMS Multiband Factor = 3, CAIPI shift = FOV/3 (*Setsompop et al., 2012*).

## Physiological Monitoring

For all the functional scans, subjects' heart rate and respiration were monitored using piezoelectric transducer on the non-dominant thumb and a respiratory belt around the upper rib cage, respectively. The physiological recordings were obtained at a sampling rate of 1000 Hz using a PowerLab physio box connected to a computer running LabChart 7 from ADInstruments.

## fMRI Analyses

### fMRI preprocessing

Anatomical images were bias-corrected using SPM (https://www.fil.ion.ucl.ac.uk/spm/) and segmented using FreeSurfer (*Fischl, 2012*). Functional runs were preprocessed with slice-timing correction, performed using FSL (http://fsl.fmrib.ox.ac.uk/fsl/fslwiki/), and motion correction, performed using AFNI software (https://afni.nimh.nih.gov/). No spatial smoothing was applied.

Because fast fMRI has distinct contributions from systemic physiological noise, including cardiac rhythms and respiration, physiological noise removal was performed on the visual stimulus and resting-state functional runs using dynamic regression adapted from RETROICOR (*Glover et al., 2000*) in runs where physiological recordings were successfully collected. In runs where physiological recordings were not successfully collected (2 runs total), physiological noise was removed using a statistical model of harmonic regression with autoregressive noise (HRAN) (*Agrawal et al., 2020*).

### Visual localizer

For each subject, one of the visual stimulus runs was used as a functional localizer to identify voxels that were significantly driven by the oscillating stimulus. A general linear model (GLM) was fit in FSL (http://fsl.fmrib.ox.ac.uk/fsl/fslwiki/) using sine and cosine basis functions with the same period as the stimulus. The F-statistic of the combined fit to both the sine and cosine basis function was transformed to a Z-score and voxels with a Z-score above 2.5 were selected for further analysis. This functional localizer was then constrained by its intersection with the anatomical definition of V1 (*Figure 2B*). Specifically, the V1 segmentation was generated automatically from the MEMPRAGE volume based on the cortical surface reconstruction generated using FreeSurfer (*Fischl, 2012*). The selected voxels from the localizer run were then mapped to each other functional run in a single transformation step. This was done by first registering all functional runs to the anatomical scan using boundary-based registration (*Greve and Fischl, 2009*) and then resampling the desired volume into the localizer field of view using the registration matrices.

## LGN segmentation

The lateral geniculate nucleus (LGN) was segmented using both anatomical and functional constraints. The anatomically defined boundaries of the LGN were generated using the FreeSurfer developmental version that generates an individual-level probabilistic atlas in individual anatomical space (*Iglesias*

*et al., 2018*). From this probabilistic atlas we considered voxels with at least a 30% probability of falling within LGN and dilated this mask to capture border voxels. We next applied a functional constraint using the visual localizer where voxels in the dilated mask with a Z-score above 2.5 were considered part of our final LGN map in each subject.

## Voxel-wise phase analysis and groupings

We then averaged the two remaining visual stimulus runs and extracted voxel's average time series between the two runs. We discarded the first 14 s to analyze the steady-state response to the visual stimulus. An estimation of the voxel's lag in relation to the oscillating stimulus was calculated using the arctangent of the sine and cosine regressor estimates. This allowed us to generate a histogram of latencies to the visual stimulus (*Figure 2C*). To extract 'fast' and 'slow' reacting voxels, a Gaussian model was fit to the histogram of phase delays. The centroid ($b$) and Full Width at Half Maximum (*FWHM*) of the Gaussian fit were calculated. Groups of fast and slow voxels were then Edges of the Gaussian fit were defined as $\pm \frac{1}{2} FWHM$ and fast and slow groups were made that each had a width of $\frac{1}{3}FWHM$. Fast voxels were identified as being within $\left[ b - \frac{FWHM}{2}, b - \frac{FWHM}{6} \right]$ while slow voxels were identified as $\left[ b + \frac{FWHM}{6}, b + \frac{FWHM}{2} \right]$ (*Figure 2C*). This procedure was done for each individual subject and on average yielded 142 fast voxels (range 64–236) and 139 slow voxels (range 68–254). Masks of fast and slow voxels were generated per subject and then mapped to the resting-state runs to inform the spectral analysis (*Figure 2D*).

## Resting-state spectral analysis

Fast and slow voxels were always identified in task-driven runs, allowing us to assess frequency content in the resting-state run using fully independent data. The maps of fast and slow voxels were registered to each individual resting-state run, and for each voxel within these masks, after discarding the first 14 s, the voxel-wise resting-state power spectrum was calculated using the Chronux toolbox (*Bokil et al., 2010*) with five tapers. We used four features to characterize the resting-state spectra: (1) slope of linear fit under 0.2 Hz; (2) the exponent of the aperiodic 1 /f fit under 0.5 Hz; (3) the amplitude of low frequency fluctuations (ALFF); and (4) the fractional ALFF (fALFF). Each of these features was z-scored within the run and then averaged on a voxel-wise basis between the two resting-state runs. All analysis of the resting-state spectra was performed in MATLAB. See *Figure 3A–D* for more information.

### Slope

A linear fit was generated for each voxel's resting-state spectrum under 0.2 Hz using least-squares to determine the coefficients of a first order polynomial. From this we were able to record the slope of that linear fit for each voxel.

### Exponent of aperiodic fit

*Equation 1* was fit using the Levenberg-Marquardt algorithm to solve non-linear least squares for each voxel's resting-state spectra. In this equation, $F$ is the independent variable and the $b$ and $x$ are the values being fit. The exponent of the resultant fit ($x$) was recorded for each voxel.

$$y = b - \log_{10}\left( F^x \right) \tag{1}$$

### Amplitude of low-frequency fluctuations (ALFF)

For each voxel, ALFF was calculated according to the method outlined in *Zou et al., 2008*. Briefly, each voxel's time series was band pass filtered between 0.01 and 0.08 Hz. Then, the voxel's time series is transformed into the frequency domain via Fast Fourier Transform (FFT). The power at each frequency is proportional to the square of the amplitude of the FFT at that frequency, and for each voxel, the ALFF value was taken as the averaged square root of the power in the 0.01–0.08 Hz frequency range. This is shown in *Equation 2* where $FFT(k)$ is the magnitude of the FFT at frequency $k$, $N$ is the number of time points, $P$ is the power spectrum, and the bar represents the average in the specified range.

$$P = \frac{|FFT|^2}{N}; \ ALFF = \sqrt{\overline{P}}_{[0.01, 0.8]} \tag{2}$$

## Fractional ALFF (fALFF)

Each voxel's fALFF was calculated as described in *Zou et al., 2008*. Fractional ALFF is briefly defined as the ratio of the power of each frequency at the low frequency range (0.01–0.08 Hz) to that of the 'global' frequency range (0.01–0.25 Hz). See *Equation 3*.

$$fALFF = \frac{ALFF_{[0.01,\ 0.8]}}{ALFF_{[0.01,0.25]}} \tag{3}$$

## Breath-hold latency calculations

Voxel-wise hemodynamic latencies were calculated according to the method outlined in *Chang et al., 2008*. Each voxel's hemodynamic latency was defined as the time-lag yielding the maximum cross-correlation between the given voxel's time series, $x(t)$, and a reference time series, $y(t)$. The reference time series was found by taking the average time series across voxels in the brain that exceeded a minimum correlation of $r > 0.25$ with the breath hold task regressor. This breath hold task regressor was defined as the convolution of a box car function, where the value is set to 1 during the breath hold and 0 at other times, and a sign-reversed canonical HRF (*Glover, 1999*). Both $x(t)$ and $y(t)$ were resampled to a resolution of 100ms before computing cross-correlations.

## Support vector machine (SVM) classification of fast, slow, and LGN voxels

SVM classifiers were trained both within and across subjects using Scikit-learn in Python (*Pedregosa, 2011*). Three models were trained with different predictive features: (1) the resting-state spectrum between 0 and 0.5 Hz, (2) the 4 features of the resting-state spectrum previously identified, and (3) the latency of the response to the breath hold task. The resting-state spectrum was extracted by taking the power at frequencies up to 0.5 Hz ultimately generating a set of 461 features. For all models, before being put into the classifier, the data was normalized by removing the mean and dividing by the standard deviation across voxels for each feature independently using the StandardScaler function of Scikit-learn. For all models, the parameters of the SVM classifier were as follows: regularization parameter ($C$)=10, kernel type =radial basis function (rbf), kernel coefficient ($\gamma$) = $\frac{1}{n_{features}*X.variance}$ . To get validation accuracies both within and across subjects, 1000 bootstraps were performed where the 80–20 test-train split of voxels was randomly chosen for each bootstrap. The average validation accuracies over the 1000 bootstraps were calculated along with the 95% confidence intervals. This methodology was followed for all 3 SVM classifiers.

## Support vector machine (SVM) regression to predict relative hemodynamic response latency

SVM models for regression were trained within subject to continuously predict the hemodynamic response latency estimated from the visual stimulus. We limited the regression to voxels whose hemodynamic response latency relative to the median was between [–3, 3] sec. Two different models were trained with different predictive features: (1) the resting-state spectrum between 0 and 0.5 Hz and (2) the 4 features of the resting-state spectrum previously identified. The input data to the model was normalized by removing the mean and dividing by the standard deviation cross voxels using the StandardScaler function of Scikit-learn. The parameters of the SVM regression model were as follows: kernel type = radial basis function, kernel coefficient ($\gamma$) = $\frac{1}{n_{features}*X.variance}$, regularization parameter = 10, epsilon = 0.1. To get the validation coefficient of determination ($R^2$, 1000 bootstraps were performed on an 80–20 test-train split of voxels that were randomly chosen for each bootstrap. The average $R^2$ over the 1000 bootstraps was calculated along with the 95% confidence interval.

## Replication dataset at 3 Tesla

To generate the replication figure (*Figure 3—figure supplement 1*), we analyzed 10 subjects (mean age = 23 years, range = 19–29, 6 female) from a previously published dataset (*Williams et al., 2023*). Subjects were selected for analysis if they had completed at least one run of a visual task as well as one resting-state scan of at least 10 min. Briefly, subjects were scanned on a 3 Tesla Siemens Prisma scanner with a 64-channel head and neck coil. Anatomical runs were acquired with 1 mm isotropic

T1-weighted multi-echo MPRAGE (*van der Kouwe et al., 2008*). Functional runs were acquired using TR = 0.378 s, 2.5 mm isotropic voxels, and multiband factor = 8. Similar preprocessing steps were performed for this dataset as described above in **fMRI Preprocessing** including slice-timing correction and motion correction. More information about the specific acquisition parameters and preprocessing can be found in the original open-access publication (*Williams et al., 2023*).

The visual stimulus was a counterphase 12 Hz flickering checkerboard stimulus that lasted 254 s with fixed 16 s ON and 16 s OFF periods beginning with an OFF period. To assist the subjects with fixation, in the center of the checkerboard was a red dot that changed brightness at random interval. Subjects were instructed to press a button whenever they detected a color change. During the resting-state scans subjects performed a behavioral task where they were instructed to close their eyes and press a button on a response box on every breath in. This behavioral task enabled us to behaviorally monitor if the subjects fell asleep. Only subjects that were consistently doing the behavioral task during the entire resting-state run were included in our analysis.

A GLM was fit in FSL to identify voxels that were driven by the stimulus and voxels with a Z-statistic above 2.5 were selected for further analysis. This functional localizer was then further constrained by its intersection with the anatomical definition of V1. Voxel-wise hemodynamic response lags within this combined mask were calculated and groups of fast and slow voxels were extracted using the methods described above in **Voxel-wise Phase Analysis and Groupings**. Then, the fast and slow voxels were registered to the resting-state run and the same procedure outlined above in **Resting-state Spectral Analysis** was used to extract the 4 spectral features of interest.

## Acknowledgements

This work was supported by National Institutes of Health grants R00-MH111748, U19-NS123717, and R01-AG070135, the Searle Scholars Program, the Pew Biomedical Scholars Program, the Sloan Fellowship, and the One Mind Rising Star award. Resources were provided by NIH grant P41-EB030006 and T32-GM008764.

## Additional information

### Funding

| Funder | Grant reference number | Author |
| --- | --- | --- |
| National Institutes of Health | R00-MH111748 | Laura D Lewis |
| National Institutes of Health | U19-NS123717 | Laura D Lewis |
| National Institutes of Health | R01-AG070135 | Laura D Lewis |
| Searle Scholars Program | | Laura D Lewis |
| Pew Charitable Trusts | Pew Biomedical Scholars Program | Laura D Lewis |
| Sloan Research Fellowship | Sloan Fellowship | Laura D Lewis |
| One Mind | One Mind Rising Star Award | Laura D Lewis |
| National Institutes of Health | T32-GM008764 | Sydney M Bailes |

The funders had no role in study design, data collection and interpretation, or the decision to submit the work for publication.

### Author contributions

Sydney M Bailes, Conceptualization, Data curation, Software, Formal analysis, Validation, Investigation, Visualization, Methodology, Writing - original draft; Daniel EP Gomez, Beverly Setzer, Software,

Investigation, Writing - review and editing; Laura D Lewis, Conceptualization, Resources, Supervision, Funding acquisition, Validation, Investigation, Visualization, Methodology, Project administration, Writing - review and editing

### Author ORCIDs
Sydney M Bailes (iD) http://orcid.org/0000-0003-1712-9233
Laura D Lewis (iD) http://orcid.org/0000-0002-4003-0277

### Ethics
All experimental procedures were approved by the Massachusetts General Hospital Institutional Review Board and all subjects provided informed consent. (protocol number: 2014P001068).

### Decision letter and Author response
Decision letter https://doi.org/10.7554/eLife.86453.sa1
Author response https://doi.org/10.7554/eLife.86453.sa2

---

## Additional files

### Supplementary files
• MDAR checklist

• Supplementary file 1. Subject-wise p-values of Wilcoxon rank-sum test comparing each spectral feature in fast and slow voxels.

• Supplementary file 2. Subject-wise average SVM classification accuracy with confidence intervals.

• Supplementary file 3. Subject-wise results from fit of linear model relating each spectral feature with phase.

• Supplementary file 4. Subject-wise performance of regression model to predict specific response timing from spectral features.

• Supplementary file 5. Parameters of simulated HRFs.

### Data availability
The data used in this paper has been deposited on OpenNeuro (https://doi.org/10.18112/openneuro.ds004645.v1.0.0).

The following dataset was generated:

| Author(s) | Year | Dataset title | Dataset URL | Database and Identifier |
|---|---|---|---|---|
| Bailes SM, Gomez DEP, Setzer B, Lewis LD | 2023 | Resting-state fMRI signals contain spectral signatures of local hemodynamic response timing | https://doi.org/10.18112/openneuro.ds004645.v1.0.0 | OpenNeuro, 10.18112/openneuro.ds004645.v1.0.0 |

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
