## [Editor Report]

This manuscript addresses the important issue of hemodynamic response function (HRF) variability across brain areas and will be valuable to researchers who use fMRI and other types of functional imaging that rely on neurovascular coupling. Using simulations and experiments, the authors provide compelling evidence that differences in the HRF can impact spectrum-based metrics such as ALFF and fALFF. A better understanding of the variability of the HRF is critical for the proper interpretation of activation onset times and of differences observed in clinical populations where both neural and vascular alterations can be expected.

---

## [Decision Letter]

**Decision letter after peer review:**

Thank you for submitting your article "Resting-state fMRI signals contain spectral signatures of local hemodynamic response timing" for consideration by *eLife*. Your article has been reviewed by 2 peer reviewers, one of whom is a member of our Board of Reviewing Editors, and the evaluation has been overseen by Timothy Behrens as the Senior Editor.

Essential revisions:

1) The authors should consider and discuss the impact of potential differences in HRF shape beyond the standard model and visual cortex. Simulations considering other variations of hemodynamic responses, especially those derived empirically, would be valuable in determining the extent to which their findings are generalizable.

2) The authors should clarify the selection of voxels for training and testing, keeping proximity in mind.

3) Additional context for the interpretation of fast vs slow responders should be given (anatomical/functional location, proximity to vasculature).

*Reviewer #2 (Recommendations for the authors):*

1. If the authors have collected any similar data with different acquisition parameters (such as 3T, lower resolution), a comparison could be helpful for understanding how well the approach translates to other imaging parameters.

2. For the SVM classification, it is mentioned that voxels are randomly sampled for the 80-20 splits. It would seem possible that certain voxels in an instance of the test set could happen to be in close proximity to voxels in the associated training set, and any smoothness in the reconstructed fMRI data might introduce bias in the performance (despite the fact that the authors have not applied spatial smoothing in post-processing). To examine this possibility, voxels in each test partition could be constrained to be at least a certain distance away from voxels in the associated training split.

3. The interpretation of the slow v. fast V1 voxel groups could be discussed. Are the fast and slow voxels found in consistent locations across different subjects, and do they map to different anatomic regions within the visual cortex (similar to the example in Figure 1E)? What might cause a voxel to belong to one or the other group – is there a relation to the underlying neural or vascular anatomy of these subjects?

4. To examine the influence of thermal noise, the authors have taken care to show that removing a fitted noise floor alters the spectral measures of interest by only a small amount (Figure S2). To further test if there is a relationship between the thermal noise content and the HRF phase, perhaps one could see if the fast versus slow V1 voxel groups have any difference in the estimated noise floor (using an appropriate normalization scheme to handle the arbitrary units of fMRI).

5. The amount of head motion in these subjects during each condition (breath-hold, rest, visual) could be reported.

---

## [Author Response]

Essential revisions:1) The authors should consider and discuss the impact of potential differences in HRF shape beyond the standard model and visual cortex. Simulations considering other variations of hemodynamic responses, especially those derived empirically, would be valuable in determining the extent to which their findings are generalizable.

We have now added new analyses exploring a range of HRF shapes to expand on this question (Supplementary Figure S2). Empirically, the double γ model does capture most HRFs typically reported in the human brain, once the duration of neural activity is accounted for and the impulse response is identified. The vast majority of fMRI studies utilize some variation of this shape of HRF in their statistical analyses, as this model captures HRF shapes that have been observed across diverse cortical and subcortical areas. Although there are studies that have investigated alternative models of the HRF (1, 2) their general shape is similar to the HRFs presented here and they have not been as widely used. Additionally, studies that have empirically derived the HRF across the cortex have presented HRFs that generally follow the shape of a double-γ function with time-to-peak (TTP) and full-width at half maximum (FWHM) values in a similar range to our simulated HRFs (3, 4). Prior work has also shown TTP and FWHM of the HRF are correlated (5, 6) informing our decision to model early onset HRFs with a shorter response duration. We acknowledge that the HRFs presented in these simulations by no means capture the full range of shapes the HRF could in principle take across the brain; however, they span a wide range of empirically derived parameters leading us to believe we are capturing a large portion of possible HRF shapes. Still, the reviewer’s point is correct that other shapes can exist, and our decision to correlate TTP with FWHM in our simulated HRFs does not allow us to disentangle the effects that TTP and FWHM have on the power spectra. To remedy this, we generated new sets of HRFs either holding the TTP or FWHM constant while the other parameter varies and created a supplementary figure to report on these additional shapes. The following text has been added to reflect this new analysis:

“And finally, although prior work has shown that the TTP and FWHM of the HRF are correlated (5, 6), we also tested the effect of holding one parameter constant (Figure 1 —figure supplement 2). Varying the FWHM had a more profound impact on the frequency spectrum compared to varying the TTP (Figure 1 —figure supplement 2), as expected due to the higher frequency content of narrower HRF shapes, although short TTPs also had a smaller effect on the spectrum.”

We have also added the following text to the discussion to discuss the reviewer’s point that these HRF shapes may vary in other cortical regions:

“One limitation of our study is that our approach has thus far only been validated in the visual system, which is the basis for the majority of our knowledge about the HRF. Given that the HRF has been shown to vary across the brain, the generalizability of our approach to different brain regions has yet to be established. However, studies that have empirically derived the HRF in other brain regions, such as the motor cortex (5) or somatosensory cortex (6), have produced HRFs with similar shapes and parameters to the double-γ HRF utilized in our simulations (3–6) providing evidence that this approach may be generalizable across the brain. Still, future work could investigate this approach in other primary sensory systems such as the auditory system where there are reliable stimulation paradigms. Additionally, this study used data collected at an ultra-high magnetic field strength (7T) which affords a higher signal-to-noise ratio (SNR) compared to more traditional 3T fMRI. However, we also performed the same spectral analysis in an independent dataset acquired at 3T and robustly replicated our result, demonstrating that this pattern is preserved despite a lower SNR and different acquisition parameters (Supplementary Figure S7).”

2) The authors should clarify the selection of voxels for training and testing, keeping proximity in mind.

We greatly appreciate the suggestion and agree that voxels in the test set being in close proximity to voxels in the associated training set potentially biases our classifier performance. We conducted new analyses to control for this possibility, requiring that our training and test voxels be in different hemispheres. Encouragingly, despite the smaller training set size in this control analysis, we still successfully decode voxel lags, and furthermore this change actually *improved* the performance of our prediction when generalizing the model across subjects. We have added the following text and supplemental figure to address this:

“We next considered whether correlated information between neighboring voxels could be contributing this prediction, to test whether we could generalize to distant voxels. To control for instances of voxels in the test set being in close proximity to voxels in the training set, we trained a new set of models using voxels in a single hemisphere (left or right) as the training set, and voxels in the other hemisphere as the testing set. Following this procedure, the number of voxels for training and testing was often closer to a 50-50 split than an 80-20 split; however, validation accuracies both within individual subjects and across subjects nevertheless remained well above chance. Interestingly, the overall performance across subjects was higher, suggesting better generalization across subjects when training on spatially separated voxels (Figure 6 —figure supplement 1).”

3) Additional context for the interpretation of fast vs slow responders should be given (anatomical/functional location, proximity to vasculature).

The following discussion paragraph has been added to address this point:

“Our results highlight the substantial variability in hemodynamic timing even within a single cortical area, and provide a way to predict this speed, but the specific reasons why individual voxels are fast or slow are not yet clear. A multitude of neurovascular and anatomical factors contribute to the variability in hemodynamic response timing across the brain. For example, the anatomical organization of the vascular network inherently introduces temporal dispersion to the hemodynamic response. As blood travels through the vascular network in response to changes in neuronal activity it not only encounters a range of vessel diameters, but also many branch points which each introduce delays. Prior work has shown that the timing and duration of the hemodynamic response both increase towards the cortical surface while the fastest, narrowest hemodynamic responses occurred in deeper gray matter (5–7). This suggests the possibility that the fast responding voxels identified in our study are more concentrated in deeper gray matter, although the spatial resolution of our images was not sufficient to separate voxels by cortical depth in this study. Vascular elasticity and reactivity also change across the vascular network and vary between regions further contributing to the variable timing of the hemodynamic response across the brain (8, 9). Arteries are surrounded by smooth muscle cells that rapidly relax and expand their diameter in response to neural activity while veins dilate much more slowly (9). Local differences in the relative concentrations of arteries and veins could therefore also introduce differences in the temporal dynamics of the BOLD signal (3, 5, 6), suggesting that the slow responding voxels may be more likely to contain veins. These relationships between vascular anatomy and timing further highlight that fMRI studies analyzing spectral power or temporal autocorrelation should take into consideration how local vascular anatomy affects these metrics.”

Reviewer #2 (Recommendations for the authors):1. If the authors have collected any similar data with different acquisition parameters (such as 3T, lower resolution), a comparison could be helpful for understanding how well the approach translates to other imaging parameters.

We appreciate the suggestion and agree that demonstrating the validity of the approach under different acquisition parameters would strengthen the generalizability of this work. Certainly, the fact that this work was done at 7T limits the generalizability of these findings given the relatively limited availability of 7T fMRI scanners. We thus investigated our approach using 3T fast-fMRI data previously collected in our lab. We found that we robustly replicated our finding in the 3T dataset, and added the following text and figures reporting this replication:

“Additionally, this study used data collected at an ultra-high magnetic field strength (7T) which affords a higher signal-to-noise ratio (SNR) compared to more traditional 3T fMRI. However, we also performed the same spectral analysis in an independent dataset acquired at 3T and robustly replicated our result, demonstrating that this pattern is preserved despite a lower SNR and different acquisition parameters (Figure 3 —figure supplement 1).”

“Replication dataset at 3 Tesla

To generate the replication figure (Figure 3 —figure supplement 1), we analyzed 10 subjects (mean age = 23 years, range = 19-29, 6 female) from a previously published dataset (10). Subjects were selected for analysis if they had completed at least one run of a visual task as well as one resting-state scan of at least 10 minutes. Briefly, subjects were scanned on a 3 Tesla Siemens Prisma scanner with a 64-channel head and neck coil. Anatomical runs were acquired with 1 mm isotropic T1-weighted multi-echo MPRAGE (11). Functional runs were acquired using TR = 0.378 s, 2.5 mm isotropic voxels, and multiband factor = 8. Similar preprocessing steps were performed for this dataset as described above in fMRI Preprocessing including slice-timing correction and motion correction. More information about the specific acquisition parameters and preprocessing can be found in the original open-access publication (10).

The visual stimulus was a counterphase 12-Hz flickering checkerboard stimulus that lasted 254 seconds with fixed 16 seconds ON and 16 seconds OFF periods beginning with an OFF period. To assist the subjects with fixation, in the center of the checkerboard was a red dot that changed brightness at random interval. Subjects were instructed to press a button whenever they detected a color change. During the resting-state scans subjects performed a behavioral task where they were instructed to close their eyes and press a button on a response box on every breath in. This behavioral task enabled us to behaviorally monitor if the subjects fell asleep. Only subjects that were consistently doing the behavioral task during the entire resting-state run were included in our analysis.

A GLM was fit in FSL to identify voxels that were driven by the stimulus and voxels with a Z-statistic above 2.5 were selected for further analysis. This functional localizer was then further constrained by its intersection with the anatomical definition of V1. Voxel-wise hemodynamic response lags within this combined mask were calculated and groups of fast and slow voxels were extracted using the methods described above in Voxel-wise Phase Analysis and Groupings. Then, the fast and slow voxels were registered to the resting-state run and the same procedure outlined above in Resting-state Spectral Analysis was used to extract the 4 spectral features of interest.”

2. For the SVM classification, it is mentioned that voxels are randomly sampled for the 80-20 splits. It would seem possible that certain voxels in an instance of the test set could happen to be in close proximity to voxels in the associated training set, and any smoothness in the reconstructed fMRI data might introduce bias in the performance (despite the fact that the authors have not applied spatial smoothing in post-processing). To examine this possibility, voxels in each test partition could be constrained to be at least a certain distance away from voxels in the associated training split.

This is now addressed in the above section “Essential Revision #2”, we have now performed the control analyses as suggested (Supplementary Figure S5).

3. The interpretation of the slow v. fast V1 voxel groups could be discussed. Are the fast and slow voxels found in consistent locations across different subjects, and do they map to different anatomic regions within the visual cortex (similar to the example in Figure 1E)? What might cause a voxel to belong to one or the other group – is there a relation to the underlying neural or vascular anatomy of these subjects?

This is now addressed in the above section “Essential Revision #3”, we have added a discussion paragraph.

4. To examine the influence of thermal noise, the authors have taken care to show that removing a fitted noise floor alters the spectral measures of interest by only a small amount (Figure S2). To further test if there is a relationship between the thermal noise content and the HRF phase, perhaps one could see if the fast versus slow V1 voxel groups have any difference in the estimated noise floor (using an appropriate normalization scheme to handle the arbitrary units of fMRI).

We appreciate the suggestion and we have now performed this analysis. We did not find a significant difference in the thermal noise floor of fast vs. slow voxels across subjects. There might be some relationship between the thermal noise content and the HRF phase, but the nature of this relationship is unclear at this point and could be investigated in future studies. We added this as a supplementary figure.

5. The amount of head motion in these subjects during each condition (breath-hold, rest, visual) could be reported.

We greatly appreciate the suggestion and have added a new analysis of motion, with the following text and figure to address this point:

“One potential factor is subject motion (Figure 5 —figure supplement 1). There was significantly more motion during the breath hold task compared to the visual stimulus (Wilcoxon rank-sum test, p = 0.01), which could lead to breath tasks providing less reliable information.”

Citations referenced:

1. M. A. Lindquist, J. Meng Loh, L. Y. Atlas, T. D. Wager, Modeling the hemodynamic response function in fMRI: Efficiency, bias and mis-modeling. *NeuroImage* 45, S187–S198 (2009).

2. M. A. Lindquist, T. D. Wager, Validity and power in hemodynamic response modeling: A comparison study and a new approach. *Human Brain Mapping* 28, 764–784 (2007).

3. A. J. Taylor, J. H. Kim, D. Ress, Characterization of the hemodynamic response function across the majority of human cerebral cortex. *NeuroImage* 173, 322–331 (2018).

4. D. A. Handwerker, J. M. Ollinger, M. D’Esposito, Variation of BOLD hemodynamic responses across subjects and brain regions and their effects on statistical analyses. *Neuroimage* 21, 1639–1651 (2004).

5. J. C. Siero, N. Petridou, H. Hoogduin, P. R. Luijten, N. F. Ramsey, Cortical Depth-Dependent Temporal Dynamics of the BOLD Response in the Human Brain. *J Cereb Blood Flow Metab* 31, 1999–2008 (2011).

6. J. A. de Zwart, *et al.*, Temporal dynamics of the BOLD fMRI impulse response. *Neuroimage* 24, 667–677 (2005).

7. H. M. Duvernoy, S. Delon, J. L. Vannson, Cortical blood vessels of the human brain. *Brain Research Bulletin* 7, 519–579 (1981).

8. K. Uludağ, P. Blinder, Linking brain vascular physiology to hemodynamic response in ultra-high field MRI. *NeuroImage* 168, 279–295 (2018).

9. P. J. Drew, Vascular and neural basis of the BOLD signal. *Current Opinion in Neurobiology* 58, 61–69 (2019).

10. S. D. Williams, *et al.*, Neural activity induced by sensory stimulation can drive large-scale cerebrospinal fluid flow during wakefulness in humans. *PLOS Biology* 21, e3002035 (2023).

11. A. J. W. van der Kouwe, T. Benner, D. H. Salat, B. Fischl, Brain morphometry with multiecho MPRAGE. *NeuroImage* 40, 559–569 (2008).